# Risk and Preventive Measures Among Older Adults in Nursing Homes in Saudi Arabia: An Exploratory Study on Falls

**DOI:** 10.3390/healthcare13030342

**Published:** 2025-02-06

**Authors:** Hmoud M. Aljarbou, Alia M. Almoajel, Mohammed M. Althomali, Khaled M. Almutairi

**Affiliations:** 1Department of Community Health Sciences, College of Applied Medical Sciences, King Saud University, Riyadh 11433, Saudi Arabia; h-m-j@windowslive.com (H.M.A.); kalmutairim@ksu.edu.sa (K.M.A.); 2Optometry Department, College of Applied Medical Sciences, King Saud University, Riyadh 11433, Saudi Arabia; malthomali@ksu.edu.sa

**Keywords:** accidental fall, older adult, fall prevention, nursing home, patient safety, risk, Saudi Arabia

## Abstract

Background: Falls among older adults are a pervasive and significant concern worldwide. A practice guideline has been developed to address the prevention of falls and their resulting consequences in hospital and long-term care settings. Aim: The study aimed to assess the fall down rate and preventive tools among older adult patients in nursing homes. Methods: A cross-sectional study was conducted on randomly selected older adult patients by using a questionnaire with the Stopping Elderly Accidents, Deaths, and Injuries (STEADI) tool in nursing homes at the Ministry of Human Resource and Social Development. Results: Among 323 older adult patients, most of them (73.1%) were male, 23.8% were from Makkah, and the age ranged from 60 to 90 years and older. Results showed that 64.8% had a psychiatric disorder, 41.8% had hypertension, 38.4% had diabetes, 38.1% had movement disorders, 11.3% had heart diseases, and 1.5% had no chronic conditions. The mean STEADI tool score was 3.5 out of 12, and of the 323 older adult patients, 51.7% had a low risk to fall and 48.3% had a high risk to fall. Of the 13 interventions used to prevent falls, the most used intervention was rehabilitative physical therapy, followed by providing patient facilities and muscle strengthening exercises. Conclusions: The level of falls was markedly low, and a significant correlation was observed between the risk of falling and the participants’ region of residence.

## 1. Introduction

The global prevalence of falls among older adults has been documented at 26.5%, and within this rate, the regions of Oceania and America exhibit the highest prevalence of falls [1]. In 2021, the Centers for Disease Control and Prevention [2] reported that falls among older adults aged over 65 years resulted in more than 38,000 fatalities, establishing it as the primary cause of injury-related deaths within this population. In Saudi Arabia, falls constitute a major public health concern for the older adult population [3]. Specifically, in the Riyadh region, 21% of older adults experience falls annually, highlighting the urgent need for targeted interventions to reduce fall risk and improve quality of care for this vulnerable population [4,5]. Fall survivors frequently encounter hip fractures and head traumas, resulting in long-term disability and a reduced quality of life [6]. A considerable percentage, varying between 20% and 30%, of falls that occur within long-term care facilities could be prevented [7]. Older adults who are institutionalized demonstrate a heightened vulnerability to experiencing falls, which can be primarily attributable to the difficulties they face in adapting to a new environment and the restrictions put on their daily routines [8].

The phenomenon of falls is a pervasive and significant concern worldwide. Every year, a substantial number of deaths, approximately 600,000, occur globally as a result of falls, thereby establishing falls as an important factor contributing to traumatic mortality. Biological abnormalities can be identified as a contributing factor to the heightened vulnerability to falls observed in the older adult population. Approximately half of the patients residing in nursing homes encounter at least one occurrence of falling yearly [9]. About 10% of geriatric patients admitted to hospital wards experience instances of falls, leading to physical injuries in more than 25% of these individuals. In addition to the physical consequences [10], falls might encompass psychological and societal implications. The fear of falling is often seen as a psychological consequence among individuals, leading to a subsequent reduction in their participation in physical and social activities [11].

The affected population mostly comprises individuals of advanced age who have a variety of health issues, making them vulnerable and fragile [12]. The notion of fragility is commonly associated with the phenomenon of aging and the occurrence of falls. According to the World Health Organization [13], institutionalized older persons have an annual fall incidence rate that varies between 30% and 50%. Additionally, 40% of these demographics experience repeated falls. A positive link has been detected between the degree of dedication to engaging in basic activities of daily living (BADL) and the probability of encountering falls. Older individuals with functional impairments are five times more likely to experience a fall than those who can do daily activities independently. The diminished capacity of persons to conduct their routine activities can lead to a deterioration in their physical strength due to a reduction in muscular involvement [14]. In addition, the preservation of motor control and balance, which are predominantly associated with cognitive changes during aging, strongly correlates with the incidence of falls. The occurrence of an overactive bladder in older adults is a notable predictor of an elevated propensity for falls. The primary reason for this correlation is commonly ascribed to the pressing necessity of immediately accessing toilet facilities. A correlation exists between insomnia and pharmacological interventions for sleep disorders and an elevated chance of experiencing falls [15]. The decrease in physical capacity observed in older adults can be ascribed to the detrimental consequences of inadequate sleep quality. Dyspnea, a common health concern among the older population, is related to multiple factors that contribute to a substantial number of falls as recorded in previous research. Older adults demonstrate a greater inclination to use healthcare services, including emergency treatment, compared with middle-aged adults [7]. This trend can be primarily attributed to higher rates of illness, chronic degenerative conditions, and impairments among the older adult population [16]. Moreover, older adults frequently ingest a considerable quantity of medications due to a range of medical ailments, leading to the emergence of polypharmacy as a noteworthy concern linked to incidents of falling [17].

The occurrence of falls is typically not attributed to a singular risk factor, but rather to the combination of chronic and acute risk factors that an individual may possess within certain settings. For example, living alone has been identified as a potential risk factor for falls, although the extent of this association may be influenced by the specific characteristics of the individual’s dwelling [9]. Another risk factor is race, where existing information from the United Kingdom and the United States indicates that individuals of Caucasian descent exhibit a higher propensity for falling compared with people of African descent in the Caribbean, Hispanics, or South Asians [18]. This finding indicates that race might have a role in the incidence of falls. Other risk factors include utilization of certain medications [7], having chronic illnesses such as arthritis, diabetes, and depression [19,20], decline in mobility [12,18,19,21,22], the psychological phenomenon known as fear of falling [12,18,19,22,23], nutritional inadequacies [20,24], cognitive impairments [22,25], and eye problems such as having cataracts and glaucoma [10,14,17]. Also, foot-related issues, such as the presence of calluses on the big toe, abnormalities in the length of toes, the occurrence of ulcers [8], malformed nails, and overall discomfort while walking, can significantly impede balance and elevate the likelihood of experiencing [11]. The proper fit of shoes is also of significant importance [6].

Due to these risk factors, various interventions have been implemented to address the high incidence of falls. Implementing prevention measures that align with the creation of safer surroundings and the development of individualized plans for integrated activities based on the health needs of older adults are of utmost importance. For instance, exercise regimens targeting strength or balance enhancement, educational initiatives [16], and certain programs, such as the Harstad injury prevention project [25], have been expanded to encompass the entire older adult population residing in a particular town or region [18]. In addition, a practice guideline has been developed to address the prevention of falls and their resulting consequences in hospital and long-term care (LTC) settings. The objective of this guideline was to offer guidance to nurses in promoting collaborative decision-making with patients, residents, and their families when choosing the most optimal fall prevention strategies [6]. Hence, there is evidence in the literature indicating that several studies identified a high incidence of fall-related risks among the older adult population. Furthermore, a high percentage of older adults at risk of falling were found to be residing in nursing homes, particularly in the Saudi Arabian context. However, these studies have not yet developed a comprehensive prevention strategy to assist decision makers and senior individuals in mitigating these risks. Therefore, the present study aimed to assess the risk of falls and prevention measures among older adult patients in Saudi nursing homes.

## 2. Materials and Methods

### 2.1. Study Design

A cross-sectional, correlational design was used in this descriptive, quantitative study.

### 2.2. Settings

The study was conducted among older adult participants in nursing homes at the Ministry of Humen Resource and Social Development.

### 2.3. Population and Sampling

A random sampling technique using the Richard Geiger equation, with a margin of error determined as 5%, a confidence level of 95%, the population, and 50% response distribution, and using sampling size calculation of more than 311, was expected to obtain the final total sample size, which equals 323. 

The older adult patients were included if (1) they were aged 60 and above and (2) they demonstrated that they were willing to participate in the study by signing the informed consent form. Participants excluded from the study included those who refused to sign the informed consent, those who had dementia, depression, and anxiety, and immobile patients.

### 2.4. Measurement

A questionnaire was developed comprising of two parts. The first part assessed demographic data, such as age, gender, chronic diseases, and preventive measures to address risk of falls, and the second part examined the risk of fall by using the 12-item Stopping Elderly Accidents, Deaths, and Injuries (STEADI) “Stay Independent” fall risk self-assessment tool [2,26,27,28].

### 2.5. Ethical Considerations

Ethical approval was acquired through the Institutional Review Board (IRB) at the School of Medicine at King Saud University (reference number: KSU-HE-23-793, dated 22 August 2023).

The researcher then completed four documents: an introduction letter, a data collection sheet, an informed consent form, and a survey. All participants had a thorough understanding of the study’s goals, procedures, and significance. A participant information page provided them with the researcher’s contact details so they could indicate interest in taking part. After the researcher made sure the participants understood what was being asked of them, they handed them an informed consent form. The participants were asked to fill out the questionnaires, and study questions were filled out after they had completed an informed consent form. Every survey was coded to protect the confidentiality of the study participants. All data were held in a secure location in the university.

### 2.6. Data Analysis

A Statistical Package for Social Sciences (SPSS), version 26, was used to analyze the collected data and test the research hypotheses. The following statistical techniques and tests were used in data analysis:Frequencies and percentages to describe demographic variables and descriptive statistical techniques, including means and standard deviations.Cronbach’s alpha reliability (a) measures the strength of the correlation and coherence between questionnaire items (b), highlights the stability of consistency with which the instrument is measuring the concept, and (c) helps to assess the ’goodness’ of a measure.Mann–Whitney, Kruskal–Wallis, and Chi-square tests were used to study the difference and association in the level of scales according to demographic characteristics.

## 3. Results

This study involved 323 patients, and most of them (73.1%) were males. About 23.8% of the participants were from Makkah, 17% from Riyadh, 15.8% from Medina, 12.4% from Unaizah, 11.8% from Taif, 9.6% from Dammam, and 9.6% from Abha. Their age ranged from 60 to 90 years and older, with a mean age of 74.1%±10.9%. The majority (41.2%) were aged from 60 to 69 years, 31.3% from 70 to 79 years, 18% from 80 to 89 years, and 9.6% from 90 years and older (Refer to Table 1).

The results showed that 64.8% of the patients had a psychiatric disorder, 41.8% had hypertension, 38.4% had diabetes, 38.1% had movement disorder, 11.3% had heart diseases, and 1.5% had no chronic illness (See Table 2).

The results showed 13 different interventions (refer to Figure 1). The most used intervention (in percentage) was Rehabilitative Physical Therapy (37.35%), followed by Providing Patient Facilities (19.28%), Muscle Strengthening Exercises (12.46%), Creating a Safe Environment (9.24%), Balance Exercises (5.22%), Walking Program (4.42%), Installing Non-Slip Surfaces (3.61%), Education and Controlling Treatment and Diet (2.81%), Providing Safety Systems (1.20%), Installing Barriers (0.80%), and Using a Cane and Training the Resident on Balance (0.40%).

The overall STEADI mean score was 3.5 out of 12. Of the 323 participants, 51.7% had a low risk of falls and 48.3% had a high risk of falls (Table 3).

As shown in Table 4, the results showed that the majority of males (55%) had lower risk than females (56%), but the association was not significant (*p* value = 0.08). In Riyadh, the majority (65%) were low risk, Unaizah (78%) were high risk, Dammam (55%) were low risk, Makkah (75%) were high risk, Medina (75%) were high risk, Taif (61%) were low risk, and Abha (65%) were high risk. This association was significant (*p* value < 0.001). The majority (54%) of those aged from 60 to 69 years were low risk, the majority (54%) of those aged from 70 to 79 years were low risk, the majority (53%) of those aged from 80 to 89 years were high risk, and the majority (58%) aged from 90 and older were high risk; this association was not significant (*p* value = 0.59). The majority (60%) of those who had no chronic diseases were high risk. The majority (53%) of those who had hypertension were low risk. The majority (52%) who had diabetes were high risk, and the majority (61%) of those who had heart diseases were low risk. The majority (63%) who had a psychiatric disorder were low risk, and those who had movement disorders (50%) were low risk. However, the associations were not significant.

Table 5 shows the Pearson correlation coefficients between demographic characteristics and chronic diseases and the risk of falls among participants. Most of the relationships were positive, with the highest correlation coefficient (Hypertension) of 0.334, followed by age group (0.215), gender (0.196), and not having chronic diseases (0.183).

## 4. Discussion

The study assessed the risks of fall down rate and preventive tools among older adult patients in nursing homes. In this study, among 323 selected older adult patients, patients who had psychiatric disorders had the highest incidence of falls. Consistent with a previous study, Iaboni and Flint [29] reported high rates of falls among patients with psychological disorders. The results showed that more than half of participants had a low risk to fall, consistent with the study by Imaginário et al. [6], who reported 3.4% of individuals experienced at least one instance of falling; meanwhile, the majority, specifically 56.6%, did not have any falls. A mean value of 1.47 ± 0.99 falls (ranging from 1 to 7) per inhabitant was computed. The risks of falls varied depending on the regions older adult patients are living in. Amer et al. [11] demonstrated that the incidence of fear of falls among senior individuals residing in nursing homes in Cairo was greater compared with those residing in the community.

Many interventions were implemented to reduce the risk of falling down, such as rehabilitative physical therapy, providing patient facilities, and muscle strengthening exercises. These interventions had successfully reduced the risks of fall among study participants. This finding is consistent with a previous review that involved a total of 108 randomized controlled trials (RCTs), encompassing 23,407 older adult participants (average age of 76 years old) residing in community settings across 25 different countries [30]. Sherrington et al. [30] indicated that exercise regimens focused primarily on balance and functional training significantly decreased the incidence of falls when compared to a sedentary control group. Additionally, Tai Chi and other exercise programs—predominantly balance and functional exercises alongside resistance training—likely contributed to a reduction in falls [30]. In another review study, the prevention exercise regimen for falls designed for older adults should have a duration of six weeks or longer [31]. Aleixo and Abrantes [31] emphasized that the exercise program should be structured around four fundamental movement categories: locomotion, level changes, pulling and pushing, and rotations. Notably, the locomotion category should be prioritized within the program, and it is essential to incorporate exercises that enhance proprioception and functional strength [31]. Similarly, Lee et al. [32] reported that the intervention strategies encompass exercise and physical therapy regimens designed to enhance balance, gait, and strength, reduction or limitation of psychoactive medication usage, addressing orthostatic hypotension, addressing foot-related issues, adjusting footwear, adapting the home environment, and providing education to patients and caregivers. Guirguis-Blake et al. [33] found three kinds of intervention that are frequently examined in academic research. These include the implementation of multifactorial therapies, which are tailored to the specific needs of individuals, and are based on a detailed and personalized evaluation of their risk of falling. A recent study conducted in Saudi Arabia reported that the Otago Exercise Program (OEP), as implemented by physical therapists, serves as an effective strategy for preventing falls among older adults [34].

The results also showed that there is no significant association between age and gender and risk of falling, although the older adult population is at a heightened risk of experiencing falls as a result of the physiological changes associated with the natural aging process. These changes include a decrease in muscle mass, as well as the potential impact of drugs and health conditions on many aspects of physical functioning, including balance, strength, eyesight, and hearing [13]. This contrasts to Salari et al.’s [1] study, where it was reported that fall-related injuries are prevalent among the older adult population and constitute a significant contributor to pain, disability, diminished autonomy, and untimely mortality. The annual incidence of falls among individuals aged 65 and above ranges from approximately 28% to 35%, with a higher prevalence of 32% to 42% observed among those aged 70 years and older.

## 5. Conclusions

The study found that the risk of falls among the older adult population in Saudi nursing homes is low based on the mean score (3.5 out of 12) for the STEADI tool. Of the participants, 51.7% were classified as having a low risk of falling, while 48.3% were classified as having a high risk of falling. Furthermore, a significant correlation was observed between the risk of falling and the participants’ region of residence. However, no significant associations were found between the risk of falling and participants’ gender, age, or presence of chronic conditions.

## Figures and Tables

**Figure 1 healthcare-13-00342-f001:**
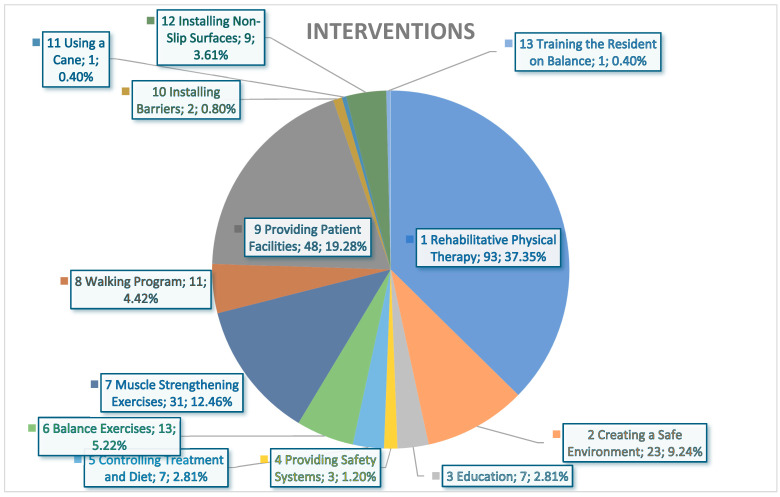
The interventions implemented among study participants.

**Table 1 healthcare-13-00342-t001:** Demographic characteristics.

Variables	Categories	N	%
Region	Riyadh	55	17
	Unaizah	40	12.4
	Dammam	31	9.6
	Makkah	77	23.8
	Medina	51	15.8
	Taif	38	11.8
	Abha	31	9.6
Age (60 to 110) Mean 74.1 ± 10.9	From 60 to 69 years	133	41.2
	From 70 to 79 years	101	31.3
	From 80 to 89 years	58	18
	90 years and older	31	9.6
Gender	Male	236	73.1
	Female	87	26.9

**Table 2 healthcare-13-00342-t002:** Chronic conditions.

Chronic Diseases	N	%
Hypertension	133	41.8
Diabetes	122	38.4
Heart Disease	36	11.3
Psychiatric disorder	206	64.8
Movement disorder	121	38.1
No chronic condition	5	1.5

**Table 3 healthcare-13-00342-t003:** Distribution of the patients according to risk level.

STEADI Score: Stay Independent	Score	Level	N	%
Ranged from 0 to 12 Mean = 3.5 ± 2.7	<4	Low risk	167	51.7
	≥4	High risk	156	48.3

**Table 4 healthcare-13-00342-t004:** Patients’ characteristics by risk level.

Characteristics		Low Risk		High Risk		$x^{2}$	*p*-Value
Gender	Male	129	55%	107	45%	3.071	0.080
	Female	38	44%	49	56%		
Region	Riyadh	36	65%	19	35%	53.638	<0.001
	Ounizah	9	23%	31	78%		
	Dammam	17	55%	14	45%		
	Makkah	58	75%	19	25%		
	Medina	13	25%	38	75%		
	Taif	23	61%	15	39%		
	Abha	11	35%	20	65%		
Age group	From 60 to 69 years	72	54%	61	46%	2.422	0.490
	From 70 to 79 years	55	54%	46	46%		
	From 80 to 89 years	27	47%	31	53%		
	90 years and older	13	42%	18	58%		
Chronic diseases	No chronic disease	2	40%	3	60%	0.279	0.598
	Hypertension	70	53%	63	47%	0.078	0.780
	Diabetes	58	48%	64	52%	1.360	0.244
	Heart Disease	22	61%	14	39%	1.436	0.231
	Psychiatric disorder	109	53%	97	47%	0.333	0.564
	Movement disorder	60	50%	61	50%	0.347	0.556

**Table 5 healthcare-13-00342-t005:** Correlations results.

Items	R	Significance
Gender	0.196 **	0.001
Region	0.162 **	0.003
Age group	0.183 **	0.001
No chronic diseases	0.183 **	0.001
Hypertension	0.334 **	0.001
Diabetes	−0.144 **	0.010
Heart Disease	−0.193 **	0.001
Psychiatric disorder	0.196 **	0.001
Movement disorder	−0.066	0.235

**. Correlation is significant at the 0.01 level (2-tailed).

## Data Availability

The data presented in this study are available on request from the corresponding author.

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
