# Peer review of "Risk and Preventive Measures Among Older Adults in Nursing Homes in Saudi Arabia: An Exploratory Study on Falls"

_healthcare, 2025, doi:10.3390/healthcare13030342_

Round 1
Reviewer 1 Report (Previous Reviewer 1)
Comments and Suggestions for Authors
The review carried out previously was complied with by the authors and the text is in accordance with the journal's parameters.
I would just like to make a small observation regarding the term "older adults" which sometimes does not have the "s" at the end (older adult). Check whether this is correct or not.
Reviewer 2 Report (Previous Reviewer 2)
Comments and Suggestions for Authors
None.
This manuscript is a resubmission of an earlier submission. The following is a list of the peer review reports and author responses from that submission.
Round 1
Reviewer 1 Report
Comments and Suggestions for Authors
I am reviewing the study "Fall Incidence and Preventive Measures among Elderly in Nursing Homes in Saudi Arabia: A Cross-sectional Study"
Suggestions:
Currently it is more common to use the term "older adults" than "elderly", I suggest the correction.
Based on the abstract, the study is exploratory, the results in percentages lead me to suggest the correction in the title to ...Saudi Arabia: An exploratory study on falls.
Abstract: "The mean STEADI tool score was 3.5, and 51.7% of the patients had low risk to fall and 48.3% had high risk to fall." I did not understand the mean was 3.5 and 51.7% for the same variable (risk of falling)? It was not clear.
The study is very well written, with current references, however item 1 (introduction) is very long. It is not common for a study in this journal to have 15 paragraphs, or practically 4 pages of introduction.
The results are also not well described, very summarized, for example, were the correlations made between sociodemographic characteristics and chronic diseases? Why? Where are the falls?
When looking at the discussion, you presented 4 small paragraphs.
The logic of an article is inverted here. It is common to make a brief introduction and a long discussion to present the findings of your work. This type of article is not the place to do a review of the literature on falls.
The study started very well, but I did not continue reading it because it got lost in the structure; it needs to be reviewed with the formats accepted by the journal and the topics "results and discussion" need to be better developed.
The subject of falls is one of the most researched among the elderly, and there are many studies to discuss.
Author Response
RESPONSE TO REVIEWER #1 COMMENTS
I am reviewing the study "Fall Incidence and Preventive Measures among Elderly in Nursing Homes in Saudi Arabia: A Cross-sectional Study"
Response to this Comment: Thank you very much for your valuable and helpful review comments.
Suggestions:
Currently it is more common to use the term "older adults" than "elderly", I suggest the correction.
Response to this Comment:
Based on the abstract, the study is exploratory, the results in percentages lead me to suggest the correction in the title to ...Saudi Arabia: An exploratory study on falls.
Response to this Comment:
Abstract: "The mean STEADI tool score was 3.5, and 51.7% of the patients had low risk to fall and 48.3% had high risk to fall." I did not understand the mean was 3.5 and 51.7% for the same variable (risk of falling)? It was not clear.
Response to this Comment:
The study is very well written, with current references, however item 1 (introduction) is very long. It is not common for a study in this journal to have 15 paragraphs, or practically 4 pages of introduction.
Response to this Comment:
The results are also not well described, very summarized, for example, were the correlations made between sociodemographic characteristics and chronic diseases? Why? Where are the falls?
Response to this Comment:
When looking at the discussion, you presented 4 small paragraphs.
Response to this Comment:
The logic of an article is inverted here. It is common to make a brief introduction and a long discussion to present the findings of your work. This type of article is not the place to do a review of the literature on falls.
Response to this Comment:
The study started very well, but I did not continue reading it because it got lost in the structure; it needs to be reviewed with the formats accepted by the journal and the topics "results and discussion" need to be better developed.
Response to this Comment:
The subject of falls is one of the most researched among the elderly, and there are many studies to discuss.
Response to this Comment:
Reviewer 2 Report
Comments and Suggestions for Authors
The present study aimed to assess the incidence of falls and prevention among elderly patients in Saudi nursing homes. However, it is not clear from the Introduction section what is the relevance and need to develop the present study. Also, after the Introduction section there is a point titled "Backgroung", which basically repeats what is in the Introduction section. Moreover, both sections are too big and are not structured in an organized way, i.e., the information does not appear in a logical way. The logical organization of the information would allow us to understand the relevance of the study, which is not verified.
Please see other comments below…
INTRODUCTION
P1L29. “On an annual basis, an estimated 1,800 older adults residing in nursing homes suffer mortality due to injuries incurred from falls (Imaginário et al., 2022).” – I believe it is important to mention that these data are for a certain population...
P2L60. "In addition, the preservation of postural control and balance, which are predominantly associated with cognitive changes during aging...” – I believe that postural control is not predominantly associated with cognition, but instead with motor control (brain stem and spinal cord - central nervous system)...
P2L78. "A practice guideline has been developed to address the prevention of falls and its resulting consequences in hospital and long-term care (LTC) settings. The objective of this guideline is to offer guidance to nurses in promoting collaborative decision-making with patients, residents, and their families when choosing the most optimal fall prevention strategies (Imaginário et al., 2022). ” – What is the relationship with the study aim?
MATERIALS AND METHODS
It is not clear in the Methods section how strategies for fall prevention were analysed...
RESULTS
The present study aimed to assess the incidence of falls... However, results on incidence are not presented... Incidence - rate of occurrence of new cases in a given period of time... Not to be confused with other parameters...
DISCUSSION
Regarding Interventions to prevent falls, exercise programs play an important role... I suggest the following two studies as important readings:
Sherrington, C.; Fairhall, N.; Wallbank, G.; Tiedemann, A.; Michaleff, Z.A.; Howard, K.; Clemson, L.; Hopewell, S.; Lamb, S.E. Exercise for preventing falls in older people living in the community. Cochrane Database Syst. Rev. 2019. [
Aleixo P, Abrantes J. Proprioceptive and strength exercise guidelines to prevent falls in the elderly related to biomechanical movement characteristics. Healthcare (Basel) 2024; 12(2): 186. http://dx.doi.org/10.3390/healthcare12020186 PMID: 38255074
Author Response
RESPONSE TO REVIEWER #2 COMMENTS
The present study aimed to assess the incidence of falls and prevention among elderly patients in Saudi nursing homes. However, it is not clear from the Introduction section what is the relevance and need to develop the present study. Also, after the Introduction section there is a point titled "Backgroung", which basically repeats what is in the Introduction section. Moreover, both sections are too big and are not structured in an organized way, i.e., the information does not appear in a logical way. The logical organization of the information would allow us to understand the relevance of the study, which is not verified.
Response to this Comment: We have revised this section extensively to provide logical organization of the information. Thank you very much.
Please see other comments below…
INTRODUCTION
P1L29. “On an annual basis, an estimated 1,800 older adults residing in nursing homes suffer mortality due to injuries incurred from falls (Imaginário et al., 2022).” – I believe it is important to mention that these data are for a certain population...
Response to this Comment: We have mentioned that in the revised version of the muscript. Thank you.
P2L60. "In addition, the preservation of postural control and balance, which are predominantly associated with cognitive changes during aging...” – I believe that postural control is not predominantly associated with cognition, but instead with motor control (brain stem and spinal cord - central nervous system)...
Response to this Comment: We have corrected this part. Thank you.
P2L78. "A practice guideline has been developed to address the prevention of falls and its resulting consequences in hospital and long-term care (LTC) settings. The objective of this guideline is to offer guidance to nurses in promoting collaborative decision-making with patients, residents, and their families when choosing the most optimal fall prevention strategies (Imaginário et al., 2022). ” – What is the relationship with the study aim?
Response to this Comment: We have corrected this part. Thank you.
MATERIALS AND METHODS
It is not clear in the Methods section how strategies for fall prevention were analysed...
Response to this Comment: We have clarified this and thank you for the valuable comment.
RESULTS
The present study aimed to assess the incidence of falls... However, results on incidence are not presented... Incidence - rate of occurrence of new cases in a given period of time... Not to be confused with other parameters...
Response to this Comment: We have reconciled this as the level of risk for falling and revised as suggested by this important comment. Based on this, we revised the title of our paper. Thank you.
DISCUSSION
Regarding Interventions to prevent falls, exercise programs play an important role... I suggest the following two studies as important readings:
Sherrington, C.; Fairhall, N.; Wallbank, G.; Tiedemann, A.; Michaleff, Z.A.; Howard, K.; Clemson, L.; Hopewell, S.; Lamb, S.E. Exercise for preventing falls in older people living in the community. Cochrane Database Syst. Rev. 2019. [
Aleixo P, Abrantes J. Proprioceptive and strength exercise guidelines to prevent falls in the elderly related to biomechanical movement characteristics. Healthcare (Basel) 2024; 12(2): 186. http://dx.doi.org/10.3390/healthcare12020186 PMID: 38255074
Response to this Comment: We have considered the suggested readings and correspondingly, added them to the references. We hope that the revised version of our work is acceptable based on your valuable comments. Again, thank you very much.